# Microstructure and Low Cycle Fatigue Properties of AA5083 H111 Friction Stir Welded Joint

**DOI:** 10.3390/ma13102381

**Published:** 2020-05-21

**Authors:** Janusz Torzewski, Krzysztof Grzelak, Marcin Wachowski, Robert Kosturek

**Affiliations:** Institute of Robots and Machines Design, Faculty of Mechanical Engineering, The Military University of Technology, 2 Kaliskiego Str., 00-908 Warsaw 49, Poland; krzysztof.grzelak@wat.edu.pl (K.G.); marcin.wachowski@wat.edu.pl (M.W.); robert.kosturek@wat.edu.pl (R.K.)

**Keywords:** friction stir welding, low cycle fatigue, 5083 aluminum alloy, fracture

## Abstract

The present paper aims to analyze the microstructure, microhardness, tensile properties, and low cycle fatigue (LCF) behavior of friction stir welded (FSW) butt joints. The material used in this study was the 5 mm thick 5083 H111 aluminum alloy sheet. Butt joints of AA 5083 H111 were manufactured at different operating parameters of the FSW process. The effect of the welding parameters on microstructure, microhardness, and tensile properties was investigated. Based on microstructure analysis and strength tests, the most favorable parameters of the FSW process were settled on the point of view of weld quality. Then, LCF tests of base material and friction stir welded specimens made of 5083 H111 were carried out for the examined welded samples under selected friction stir welding parameters. The process of low-cycle fatigue of 5083 H111 aluminum alloy was characterized by cyclic hardening for both: base material and FSW joint. It was revealed by a decrease in the width of the hysteresis loop with the simultaneous significant increase in the values of the range of stress. It was determined that fatigue cracks are initiated by cyclic slip deformation due to local stress concentration from the surface in the corner of the samples for the base material and the heat-affected zone for FSW joints. For all tested strain amplitudes, the fatigue crack propagation region is characterized by the presence of fatigue striation with secondary cracks.

## 1. Introduction

Aluminum alloys are widely used materials that play a very important role as a construction material in many industries, including the shipbuilding industry. Their application to ship hull plating and structures is increasing as the alloys make it possible to decrease mass of structures as compared with that of steel structures. Among weldable Al-alloys suitable to plastic working, the 5xxx- series (the group of Al–Mg alloys) of high strength, good weldability, and relatively good service conditions are still very popular. In particular, 5083 aluminum alloy is one typical 5xxx series aluminum alloy, in which Mg is the main alloying element to improve the corrosion resistance. Exceptional corrosion resistance of 5083-H111 is widely known and has also been confirmed in the deep-sea environment [1]. It should also be noted that in complex structures of marine vessel construction, several connections are an important issue due to the strength and durability of the structure [2,3,4]. The most common method of joining is welding, which in the case of aluminum has limitations due to welding process problems such as oxide removal and reduced strength in the weld and heat-affected zone (HAZ). Different welding methods are currently used to produce aluminum ship structures, namely, gas welding (GMAW), laser welding, and friction stir welding (FSW). Of these, FSW was also considered to be a very attractive method of joining aluminum structures due to many excellent features, such as outstanding connection parameters, a low degree of initial imperfections, and low energy consumption [5]. Besides, it is necessary that new designs and all the stages of production comply with strict environmental regulations. Within this context, the application of FSW as a manufacturing process to welding aluminum vessel structures can contribute to the development of high-speed craft and lightweight ships that are more fuel efficient [6]. A comparison of energy consumption and environmental impact of FSW and GMAW was made in the work of Shrivastava et al. [7]. For the welding parameters used in this study, joining by FSW consumes 42% less energy as compared to GMAW and utilizes approximately 10% less material for the design criteria of similar maximum tensile force. This leads to approximately 31% of less greenhouse gas emissions for FSW as compared to GMAW. Maggiolino et al. [8] compared the corrosion resistance of AA6060T5 and AA6082T6 jointed surfaces via friction stir welding (FSW) and metal inert gas (MIG) methods, respectively. This research clearly indicates higher corrosion resistance in an acid salt solution of connections made using the FSW method. The comparison of different methods of joining an aluminum alloy in the context of fatigue strength was made by Ericsson at al. [9]. That publication stated that the fatigue strength of FSW joints for Al–Mg–Si alloy was higher than that of MIG-pulse and TIG welds of the same material and that the mechanical and fatigue properties of the FSW joints were relatively independent of welding speed in the range of low to high commercial welding speed in this alloy. 

Mechanical properties and microstructure of AA5083 FSW joints with different welding process parameters have been widely studied in the literature [10,11,12]. Zhou et al. [13] studied the influence of the kissing bond on the mechanical properties and fracture behavior of AA5083-H112 friction stir welds, concluding that the welding parameters had a substantial effect on the length of the kissing bond, which was found to decrease with the increase in the welding heat input, as estimated based on the rotation and feed speeds. During their service life, marine structures tend to be subjected to repeated impulsive pressure loads that unavoidably cause the fatigue and cyclic deformation since they are subjected to cyclic stresses and strains. The applications of Al alloys and their connections in the shipbuilding industry involve checking the performance including cyclically variable loads [14]. Therefore, cyclic deformation characteristics and fatigue of the friction stir welded joints must be ascertained to guarantee the structural integrity and safe use, e.g., under the influence of impulsive pressure loads due to slamming. There have been some reports on the fatigue properties of FSW joints of heat treated [15] or numerically analyzed [16] aluminum alloys. However, previous studies focused mainly on the fatigue life [17,18] and crack growth [19,20,21], with only limited studies on the low-cycle fatigue (LCF) behavior of FSW joining in aluminum alloys [22,23,24]. To the authors’ knowledge, although the strain-hardened aluminum alloys are readily weldable, FSW resulted in good tensile properties, which must be verified concerning cyclic loading with high strain amplitude. In this study, microstructural behavior, microhardness, tensile properties, low cycle fatigue (LCF), and fracture of friction stir welded butt joints of 5083-H111 sheets were investigated. In the first stage of experimental research, the best welding parameters were established based on microstructure observation, microhardness, and analysis of tensile properties. The main part of this work focuses on the understanding of the LCF behavior of AA5083-H111 base material and weldments, made from selected parameters of friction stir welding (FSW) process. The LCF fracture surface microstructures were characterized on selected fractured specimens, and thus, the morphologies of fracture surfaces under various low cycle fatigue conditions are also reported using standard microscopic techniques.

## 2. Materials and Methods

In this study, the 5083-H111 aluminum alloy was used as the base material. The research was carried out on plates with 5 mm thickness, obtained according to the rolling direction with the following dimensions: length × width × height = 500 mm × 100 mm × 5 mm. The AA 5083-H111 is a non-heat treatable alloy, and it is only strengthened by strain hardening due to cold forming. The H111 condition was obtained with some work hardening by shaping processes but less than that is required for an H11 temper. To achieve H11 temper, the metal is strain hardened to a strength that is roughly one-eighth of the way between annealed (O) and full hard (H18). The chemical compositions and principal mechanical properties of the AA5083-H111 alloy are, respectively, presented in Table 1 and Table 2.

Friction-stir-welded butt joints were fabricated with the position-controlled FSW machine (LEGIO 4UT, ESAB, Warsaw, Poland). The FSW tool geometry consisted of a threaded conical pin (diameters from 6.5 to 8.7 mm and height of 4.8 mm) and a spiral shoulder with a diameter of 19 mm. The FSW direction was parallel to the rolling direction of the sheet. The sheets were clamped rigidly onto the top of a steel backing plate. Welding trials were carried out, and optimized process parameters presented in Table 3 were used to fabricate the weld joints free of volumetric defect (Figure 1) and lack of severe penetration for further investigation.

For microstructure analysis, samples were cut perpendicular to the welding direction (cross-welds) (Figure 2) then prepared according to the standard metallographic techniques for specimen preparation. Their microstructure has been revealed using modified Graff-Sargent reagent (3 g CrO_3_ + 87.5 mL H_2_O + 15 mL 63% HNO_3_ + 1 mL 40% HF) with etching time of about 2 min. The microstructure analysis has been performed using a digital light microscope Olympus LEXT OLS 4100 (Warsaw, Poland) equipped with MountainsMap7 software allowing grain size measurements. The Vickers microhardness of selected welds were measured on the cross-section of polished samples by applying a load of 0.98 N. The distribution of microhardness was performed for the upper, middle, and lower part of the cross-weld: 0.5, 2.5, and 4.5 mm, respectively. For mechanical testing, cross-weld flat dog-bone tensile and LCF specimens were extracted from the FSW joints as shown in Figure 2. 

Tensile tests were performed on samples with a gauge width of 15 mm and length of 60 mm according to ISO 6892-1:2019 by an INSTRON 8802 servo-hydraulic test machine at a constant displacement rate of 4 mm × min^−1^, with an extensometer having a gauge length of 50 mm. The LCF tests were carried out on specimens with a gauge width of 10 mm and length of 25 mm according to the ISO 12106:2017 in a fully reversed total strain-controlled condition (R= −1). The strain function exhibited sine waveforms and the test was conducted at a constant strain rate of 5 × 10^−4^ s^−1^ at ambient temperature. The strain amplitudes of 0.35%, 0.4%, 0.5%, and 0.6% were used for both the BM and butt-weld samples. An axial clip extensometer with a gauge length of 25 mm was used to control the amount of applied strain, and the fatigue failure criteria were defined when the maximum tensile stress dropped by 20% below that at initial life.

## 3. Results and Discussion

### 3.1. Microstructural Behavior

An example of the macrostructure of the sample A5_5-2 FSW joint of AA5083 is shown in Figure 3. 

This joint was prepared under a tool rotational speed of 500 rpm and a transverse welding speed of 200 mm/min. Figure 3 shows the typical macrostructure for friction stir welding process consisting of stir zone (SZ), thermomechanically affected zone (TMAZ), heat-affected zone (HAZ), and base material (BM). Macroscopic observations did not reveal any noticeable imperfections in the investigated joint. A worth noting fact is an excellent symmetry of the obtained joint. The significant grain refinement can be presented by comparison of BM and SZ microstructure (Figure 4a,b).

The base material microstructure has a grain size of about 30–50 μm (Figure 4a). At the same time, the far more homogeneous microstructure of SZ is characterized by grain size of about 8–10 μm (Figure 4b). The boundary between TMAZ and SZ can be observed in Figure 5a. The TMAZ is characterized by deformed, elongated grains reflecting the direction of material flow in the welding process.

Affecting of the tool on the workpiece results in the formation of ultrafine grain microstructure in the SZ due to the dynamic recrystallization phenomenon (Figure 5b). 

### 3.2. Microhardness

The results of Vickers microhardness tests on the cross-section of connections for three different measurement lines, upper, middle, and lower, are shown in Figure 6. Microhardness measurements were made at 0.5 mm (upper), 2.5 mm (middle), and 4.5 mm (lower) from the top of the sample, symbolically marked with the dotted lines in Figure 6. The hardness profile exhibits asymmetric distribution with an unusual shape for all lines, different from that are mostly observed for aluminum FSW joints [11,25] where a constant microhardness was obtained in the entire weld cross-section. Regardless of the position of the measuring line, higher microhardness values were observed in the SZ area compared to the microhardness value of about 82 HV_0.1_ for BM. The microhardness measurements depending on the position of the measuring line differed little for the BM and the HAZ. In the TMAZ, the highest microhardness of 88 HV_0.1_ for the upper line and the lowest of 85 HV_0.1_ for the lower line was observed. These results confirm the positive influence of grain refinement in the SZ area observed in Figure 4 and Figure 5 on the strength properties of aluminum alloy 5083 H111.

Though microhardness increase in friction stir welded aluminum alloys was rare, Mishra and Ma [5] referred a study showing a slight increase in hardness across the weld nugget compared to the TMAZ for a slightly (less than 6%) hardened 5083-H112 as a result of the very fine grain size formed by FSW.

### 3.3. Tensile Properties

A comparison of the average tensile properties of the base material AA5083 H111 and welded joints is shown in Figure 7. The graph presents a comparison of ultimate tensile strength (R_m_), 0.2% yield strength (R_p02_), and percent elongation at fracture (A) of data from the material standard 5083 and tests performed for the base material and FSW joints. The dash lines in Figure 7 show the minimum values strength properties described in the material standards and shown in Table 2. The colors of the dash lines correspond to the characteristic material parameters. Ultimate tensile strength results for all samples come close to 300 MPa (Figure 7) and even exceed it. The maximum value of R_m_ = 310 MPa was obtained for the base material, and the best FSW connection marked A5_5-2 (500 rpm—200 mm/min) reached 305 MPa, which is almost 98% of the strength of the base material and exceeds the strength expected in the material standard marked by the blue dashed line (Figure 7). Very promising results were also obtained with the yield strength for which the maximum stress values of 150 MPa of the FSW A5_5-2 joint constitute 90% of the strength of the base material, and the results are about 15% higher than those obtained by the GMAW and GTAW method reported by Liu et al. [26]. There was no significant decrease in elongation, only about 10% for all cases of FSW joints differently than presented by Prabha et al. [27] where the decrease in elongation reached about 70%.

In tensile tests, specimens’ failures were obtained outside the area of the joint in the heat-affected zone. This confirms the high values of strength properties and proves the good quality of FSW joints, which are promising in the industrial production of elements.

### 3.4. Fatigue Behavior

Microhardness and tensile tests were used to select technological parameters for preparing samples for low cycle fatigue (LCF) tests. The samples welded at 500 rpm tool rotation speed and 200 mm/min tool traverse speed was used for LCF tests. The process of low-cycle fatigue of 5083 H111 aluminum alloy was characterized by cyclic hardening (see Figure 8). 

It was revealed by a decrease in the width of the hysteresis loop shown in Figure 8, with the simultaneous sharp increase in the values of the range of stress for both: base material (Figure 8a) and FSW joint (Figure 8b).The observed effect of the increase in stress range under the influence of strain for the AA5083 alloy can be explained by the increase in dislocation density and the mechanism of solid solution operation and the presence of a fine second phase (Al6Mn) recognized in the literature [28]. 

Figure 9 representing a stress amplitude, σ_a_, versus plastic strain amplitude, ε_ap_/2, in bilogarithmic coordinates results in a linear curve. The correlation line describes Morrow’s Equation (1), which is analytical dependence between stress σ_a_ and plastic strain ε_ap_/2.
(1)$$log\sigma_{a}=logK^{\prime }+\;n^{\prime }log\varepsilon_{ap}$$

In Equation (1), K’ is the cyclic hardening coefficient and n’ is the cyclic hardening exponent. In the case of the lack of an explicit stabilization period of cyclic properties, the values of hysteresis loop parameters (σ_a_, ε_ap_), present in Equation (1), were identified from the point corresponding to half the fatigue life (N/N_f_ = 0.5). Both material factors were calculated on the basis of the experimental data by employing the least square method (Figure 9).

The resulting curves are characterized by the parameters listed in the table in Figure 9 which indicate a reduction of around 20% strain curve coefficient K’ and a similar change in strain curve exponent n’. This sort of change causes a decrease in the stress amplitude for FSW joints, necessary for obtaining the assumed level of plastic strain amplitude. Using the strain-life approach, the low cycle fatigue properties can be estimated by resolving the elastic and plastic strain amplitude component from the total strain amplitude. According to this approach, the fatigue life of the examined AA5083H111 was described with the equation of Manson–Coffin–Basquin (MCB) as follows (Equation (2)):
$$\frac{\Delta \varepsilon_{c}}{2}=\frac{\Delta \varepsilon_{e}}{2}+\frac{\Delta \varepsilon_{p}}{2}=\frac{\sigma_{f}^{\prime }}{E}{\left(2N_{f}\right)}^{b}+{\;\varepsilon }_{f}^{\prime }{\left(2N_{f}\right)}^{c}$$where σ’_f_ is the fatigue strength coefficient, E is Young’s modulus, b is the fatigue strength exponent, ε’_f_ is the fatigue ductility coefficient, and c is fatigue ductility exponent. These can be computed by fitting the test data to MCB Equation (2) using a series of regression analyses based on the least-squares approach. The low cycle fatigue properties obtained for the BM and FSW AA5083 samples under the range of the tested parameters are summarized in Table 4.

Table 4. shows that at the levels of total strain ε_ac_ applied in fatigue tests, the process of the cyclic strain of the examined aluminum alloy for both BM and FSW joints ran with the dominant role of plastic strain component ε_ap_. Therefore, it can be assumed that for these levels of strain ε_ac_, the cyclic strain resistance of the investigated 5083 H111 will mostly depend on its plastic properties. Analysis of the achieved fatigue graphs (Figure 10) allows to state that the joining significantly decreases low cycle fatigue life of AA5083.

The influence of friction stir welds on the service life of the tested aluminum alloy was clear at all levels of strain ε_ac_. The decrease in durability was up to 66%, and it indicates the sensitivity of the material working under low cycle loads under the presence of joints in the load area. In the event of an increase in loads above the yield point, components connected by the FSW method may be exposed to a significant deterioration in performance differently than it is presented for monotonic tensile loading referred to as joint efficiency. FSW joint efficiency exceeding 90% for this alloy is found based on both: own research (Figure 7) and literature [29].

### 3.5. Fractography

All tested LCF samples were damaged within the measuring section of the extensometer and in the case of FSW joints, within the heat-affected zone outside the FSW joint. The overall morphologies of fracture surfaces of LCF samples are shown in Figure 11 to gain insight into the effect of strain amplitudes and FSW joining on LCF failure features. It is visible that the fracture surfaces all consist of fatigue crack initiation region (as marked by arrows), fatigue crack propagation area (as surrounded by the dashed line), and fast fracture region at both base material and FSW joints for different strain amplitudes. The LCF cracks in the base material initiate from the surface in the corner of the samples (see Figure 11a,b), while the fatigue crack propagation area is quarter elliptical in shape. The crack initiation sites are flat, without any noticeable defects for both the base material and the FSW joints. These flat areas at the crack initiation sites suggest that fatigue cracking was initiated by cyclic slip deformation due to local stress concentration.

There is a slight difference for FSW joints, for which cracks initiate from the surface but the shape of the fatigue crack assumes an almost rectangular shape (see Figure 11c,d). In addition to the shape, it should be noted that the area of fatigue cracking of the base material is significantly smaller than that of welded specimens. This shows faster crack initiation and easier fatigue crack propagation in HAZ of welded specimens, which was also visible in the shorter fatigue life of the FSW joined samples as shown in Figure 10. More details about crack characteristics can be obtained based on microfractographic analysis of fractures using a SEM microscope at higher magnification.

The morphologies of fatigue crack propagation regions for two total strain levels, 0. 35% and 0. 5%, for base material and FSW connections are shown, respectively, in Figure 12 and Figure 13. Observations and analyzes were carried out for both base material and FSW joint samples analyzed at two magnifications. The analyzed area was marked with a yellow frame and shown at a larger magnification (2500×). Fatigue striations and secondary cracks are observed in the crack propagation region for all strain amplitudes and both types of samples. According to the formation mechanism of fatigue striation, described for example by Shyam et al. [30], the fatigue striation space can reflect the rate of fatigue crack propagation. Alatorre et al. reported that the wider striation is related to faster crack propagation [31]. At the strain amplitude of 0.35%, it is seen from Figure 12b that the striation spacing is narrower than in Figure 12d, which implies a low crack propagation rate. The crack propagation procedure occupies plenty of cycles of LCF life. Hence, the fatigue life of 0.35% strain amplitude is remarkably longer than that of higher strain amplitude. At the 0.5% strain amplitude (Figure 12d), the striation spacing is larger than that at 0.35% strain amplitude but still relatively narrow in comparison to crack propagation regions for FSW joints, which leads to a decreased LCF life. At FSW joints, the wide fatigue striation spacing is observed (Figure 13b) for 0.35% total strain and thus, the relatively faster crack propagation yields a lower fatigue life (see Figure 10) in comparison to the base material.

Striations observed in the crack propagation regions on both figures are accompanied by secondary cracks, indicated by green arrows. Concerning the secondary crack, its quantity is found to be greater for base material (Figure 12) than for FSW joints (Figure 13).

## 4. Conclusions

The performed research on microstructure, microhardness, mechanical properties, and low cycle fatigue behavior of friction stir welded (FSW) butt joints made of 5083 H111 aluminum alloy allowed the following conclusions to be drawn:
Macroscopic observations did not reveal any noticeable imperfections in the investigated joint. On the other hand, microscopic examinations revealed the impact of the tool on the workpiece in the friction stir zone; it results in the formation of ultrafine grain microstructure in this area due to the dynamic recrystallization phenomenon. As a result of the dynamic impact of the tool and changes in the microstructure, the microhardness of the joined material value of 82 HV0.1 increased to about 88 HV0.1 in the stir zone for selected friction stir welding parameters.It was found that the FSW joined elements subjected to low cycle fatigue may be exposed to significant efficiency deterioration reaching 66% differently than in the case of monotonic tensile load, for which the efficiency of the FSW weld exceeded 90% for this alloy both based on own research and literature. The process of low-cycle fatigue of 5083 H111 aluminum alloy was characterized by cyclic hardening. It was revealed by a decrease in the width of hysteresis loops during the fatigue test, with the simultaneous sharp increase in the values of the range of stress for both: base material and FSW joint.Failures in FSW joined specimens occurred in the heat-affected zone for both tensile and LCF testing. The scanning electron microscopy observations of the fatigue crack propagation region revealed the presence of fatigue striations, together with some secondary cracks for all strain amplitudes and both types of samples.

## Figures and Tables

**Figure 1 materials-13-02381-f001:**
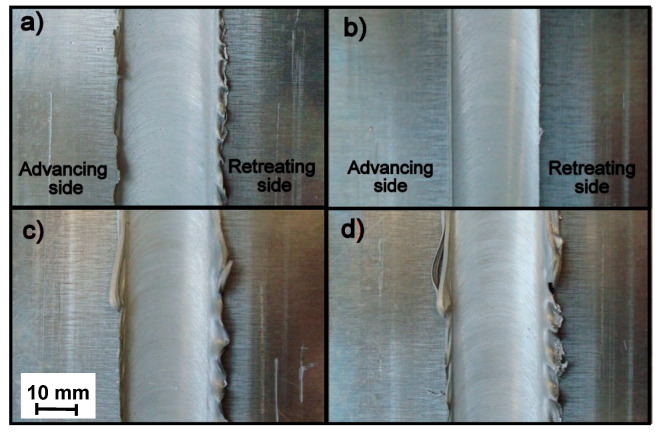
Surface morphologies of friction stir welded (FSW) joints from the face side with different connection process parameters: (**a**) A5_5-1, (**b**) A5_5-2, (**c**) A5_9-1, and (**d**) A5_9-2.

**Figure 2 materials-13-02381-f002:**
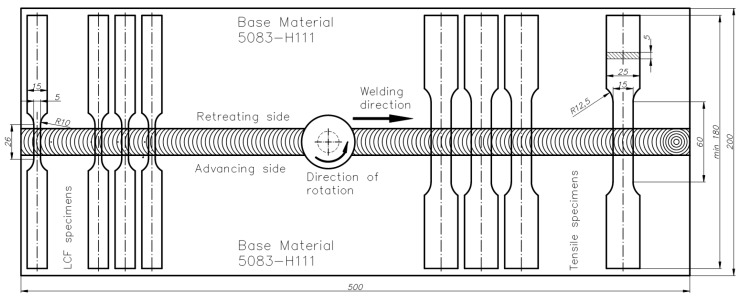
Arrangement of AA 5083-H111 sheets at the time of making FSW joints and dimensions of specimens used for tensile and low cycle fatigue (LCF) tests.

**Figure 3 materials-13-02381-f003:**
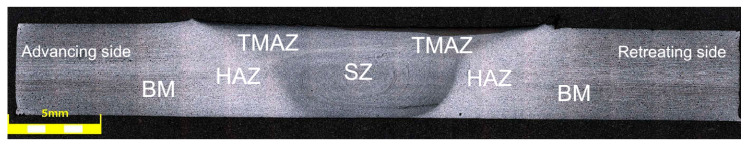
The cross-weld macrostructure of the 5083 FSW weld.

**Figure 4 materials-13-02381-f004:**
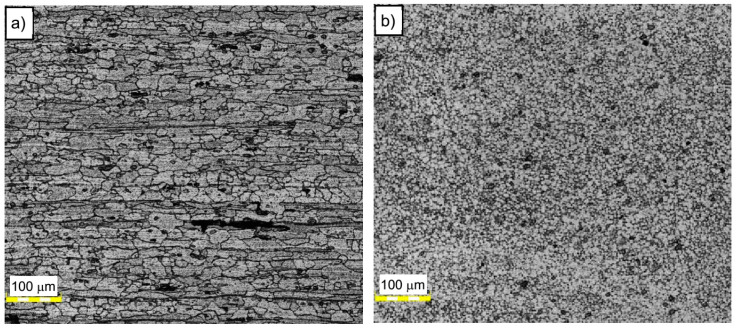
Microstructure of (**a**) base material AA5083 and (**b**) stir zone (SZ) in the A5_5-2 sample.

**Figure 5 materials-13-02381-f005:**
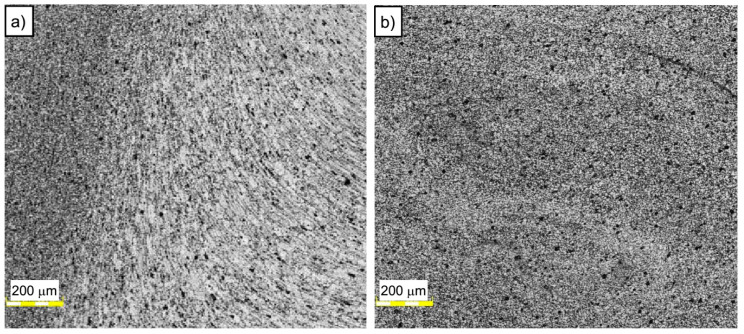
Microstructure of (**a**) boundary between thermomechanically affected zone (TMAZ) and SZ and (**b**) SZ in A5_5-2 sample.

**Figure 6 materials-13-02381-f006:**
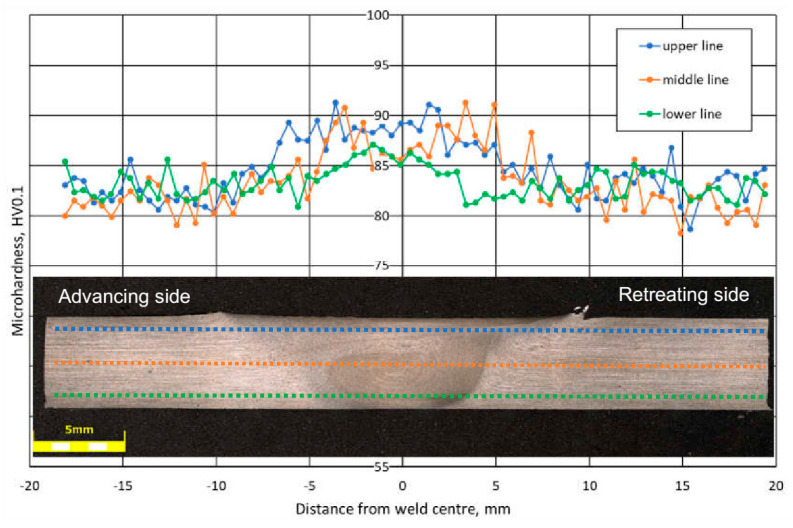
Microhardness values of AA5083 H111 FSW joint in the A5_5-2 sample.

**Figure 7 materials-13-02381-f007:**
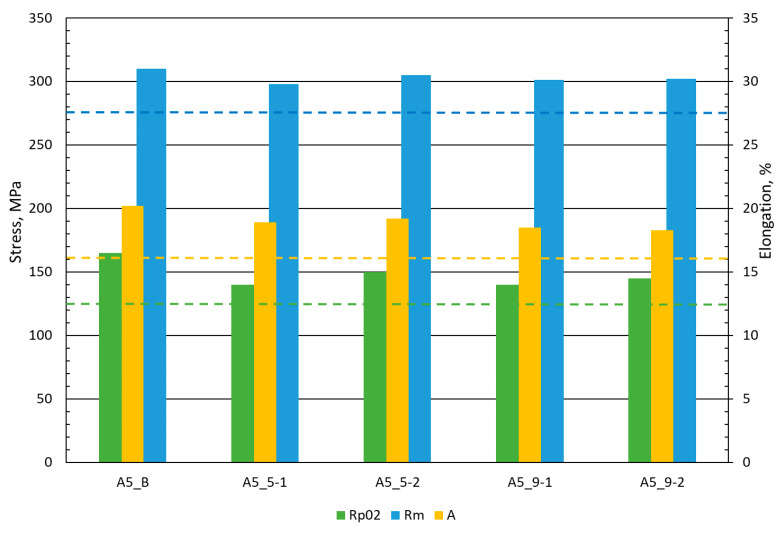
Tensile properties on the base of the AA 5083 H111 research performed for the base material (A5_B) and FSW joints (A5_5-1, A5_5-2, A5_9-1, and A5_9-2).

**Figure 8 materials-13-02381-f008:**
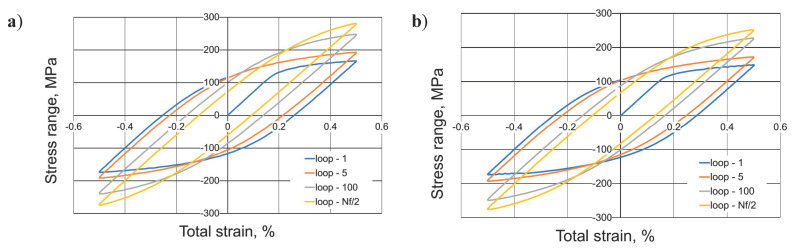
Examples of hysteresis loops for 0.5% total strain range: base material (**a**) and FSW joint (**b**).

**Figure 9 materials-13-02381-f009:**
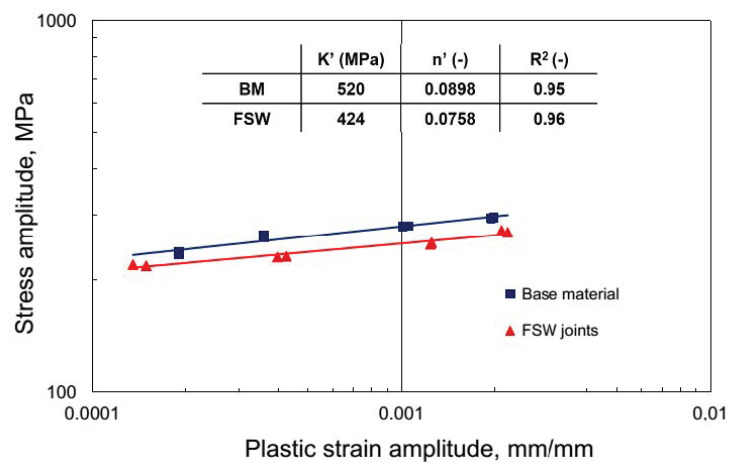
Stress amplitude versus plastic strain amplitude behavior.

**Figure 10 materials-13-02381-f010:**
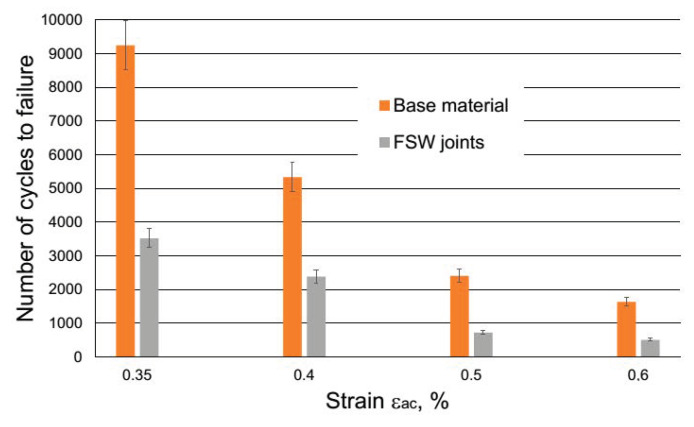
Low cycle fatigue life of AA5083 H11.

**Figure 11 materials-13-02381-f011:**
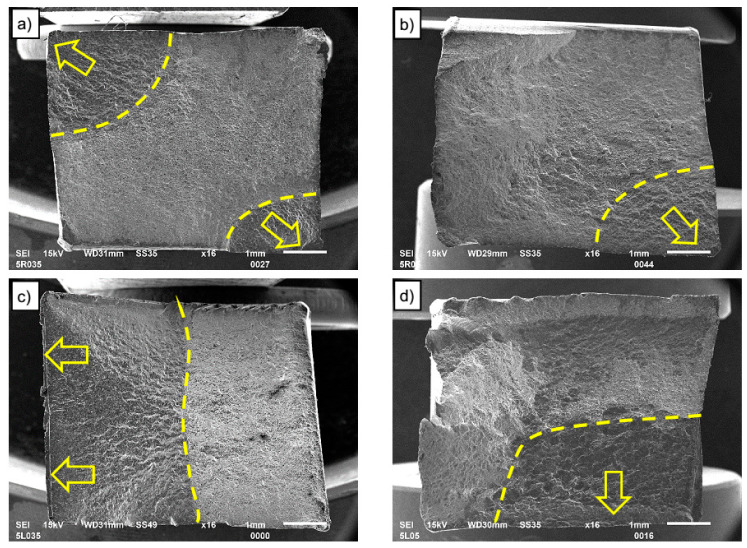
Example morphologies of fracture surfaces of base material (**a**) ε_ac_ = 0.35%, (**b**) ε_ac_ = 0.5%) and FSW joints (**c**) ε_ac_ = 0.35%, and (**d**) ε_ac_ = 0.5%).

**Figure 12 materials-13-02381-f012:**
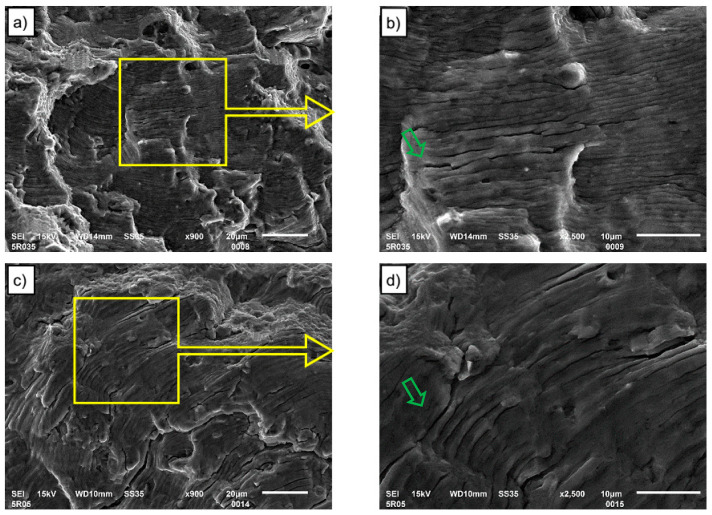
Fatigue crack propagation regions for base material at different strain amplitudes: (**a**) and (**b**)—ε_ac_ = 0.35% and (**c**) and (**d**)—ε_ac_ = 0.5%.

**Figure 13 materials-13-02381-f013:**
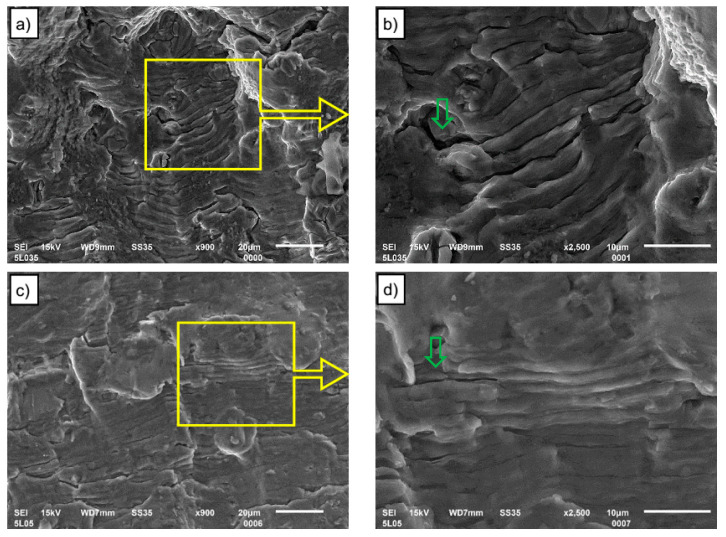
Fatigue crack propagation regions for FSW joints at different strain amplitudes: (**a**) and (**b**)—ε_ac_ = 0.35% and (**c**) and (**d**)—ε_ac_ = 0.5%.

**Table 1 materials-13-02381-t001:** Chemical composition of the 5083-H111 aluminum alloy (wt. %).

Al	Mg	Mn	Si	Fe	Cr	Zn	Ti	Cu	Other
Bal.	4.0–4.9	0.4–1.0	max 0.4	max 0.4	0.05–0.25	max 0.25	max 0.15	max 0.1	max 0.15

**Table 2 materials-13-02381-t002:** Mechanical properties of AA5083-H111 according to the standard and own research.

AA5083-H111	R_m_ (MPa)	R_p0_2 (MPa)	A (%)	HV_0.1_
**Standard**	min. 275	min. 125	16	82
**Research**	310	165	20	83

**Table 3 materials-13-02381-t003:** Summary of Welding Matrix.

		Tool Rotation Speed (rpm)	
		500	900
**Tool traverse speed (mm/min)**	100	A5_5-1	A5_9-1
	200	A5_5-2	A5_9-2

**Table 4 materials-13-02381-t004:** Estimated strain-life relationship parameters.

AA5083-H111	σ’_f_ (MPa)	b (-)	ε’_f_ (-)	c (-)
**BM**	532	−0.0859	9.445	−1.1795
**FSW joints**	682	−0.1312	2.096	−1.1312

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
