# Peer review of "Microstructure and Low Cycle Fatigue Properties of AA5083 H111 Friction Stir Welded Joint"

_materials, 2020, doi:10.3390/ma13102381_

Round 1
Reviewer 1 Report
1. What is specimen surface in line 21-22. Be specific
2. Introduction need revision. For example, several location “corrosion” term is mentioned which
is not related to current study.
3. Line 142 and 143 repetition
4. Check the grammar and sentence. For example, Correct sentence 146 and 147
5. Base metal picture (Figure 5) should come before Figure 4
6. Sentence 153 and 154: How the grain size was measured? The grain size looks smaller to me
that the reported value?
7. Line 158: check the sentence. What is top of sample?
8. Line 162: what is hardening here?
9. Line 165-167: check the sentence
10. What does mean positive influence in line 166?
11. Show the position of hardness line in the macrograph of figure 6
12. What is hardness growth in line 170?
13. Line 178: Oscillate is not appropriate word
14. Line 180: Native material should be base material. Check throughout the manuscript
15. Line 185: use the author name for referencing instead of Ref
16. Line 190-191: Check the sentence. Did the fracture occur in HAZ?
17. Line 200-201 is repetition of line 197-199
18. Line 254: how the crack initiation looks like?
19. What is difference between crack propagation and fast fracture region? Crack initiation is more
influential? Why it was not discussed in the paper?
20. Line 292: Indicate secondary crack by arrow
21. Conclusions need to be shorten and include only important obseration
22. Conclusions 5 is not clear (line 313-315)
Author Response
Dear Sir or Madam
I am sending you the revised version of the paper that we submitted to your Journal “Materials”.
The authors acknowledge both the editor and the reviewer for carefully reading the paper. The comments are very interesting and useful and give a noticeable contribution to improve the paper.
We tried to satisfy all the suggestions of the reviewers.
In the manuscript, the main changes are highlighted in red. Please, note that, for the sake of clarity, the deleted text is not reported.
I am sending you the responses to reviewers’ comments
1. What is specimen surface in line 21-22. Be specific
Author’s Response: The authors in this sentence emphasize that the cracks did not initiate from specific places such as defects or discontinuities of the weld.
2. Introduction need revision. For example, several location “corrosion” term is mentioned which is
not related to current study.
Author’s Response: The term „corrosion" appears in the first paragraph of the introduction to present
a wider perspective on the problem and to demonstrate the specific characteristics of the tested material as well as the validity of using FSW technology for joining the tested material.
3. Line 142 and 143 repetition
Author’s Response: Thank you for this comment, the authors corrected the mistake.
4. Check the grammar and sentence. For example, Correct sentence 146 and 147
Author’s Response: Thank you again, the authors corrected the sentence.
5. Base metal picture (Figure 5) should come before Figure 4
Author’s Response: Thank you for this comment, the authors have changed the order of the figures.
6. Sentence 153 and 154: How the grain size was measured? The grain size looks smaller to methat the
reported value?
Author’s Response: The grain size has been measured using MountainsMap7 software. The appropriate sentence has been added to the “Materials and Methods” part of the manuscript.
7. Line 158: check the sentence. What is top of sample?
Author’s Response: The top of the sample refers to the tool interaction side, additional lines were introduced in Fig. 6.
8. Line 162: what is hardening here?
Author’s Response: The used word was not appropriate and the sentence was rebuilt in the new version of the manuscript.
9. Line 165-167: check the sentence10. What does mean positive influence in line 166
Author’s Response: The sentence was rebuilt. The positive influence in this sentence relates to the increase in microhardness in the joint area as a result of tool interaction.
11. Show the position of hardness line in the macrograph of figure 6
Author’s Response: Thank you for this comment, the authors added the microhardness line designation in photo 6 and an additional description in the article.
12. What is hardness growth in line 170?
Author’s Response: Thank you again, in this sentence there should be „microhardness increase”
13. Line 178: Oscillate is not appropriate word
Author’s Response: The authors have rewritten the sentence.
14. Line 180: Native material should be base material. Check throughout the manuscript
Author’s Response: Thank you for this comment, the authors have introduced the same description of
material throughout the article
15. Line 185: use the author name for referencing instead of Ref
Author’s Response: Thank you for this comment, the authors have introduced an amendment.
16. Line 190-191: Check the sentence. Did the fracture occur in HAZ?
Author’s Response: Thank you for this comment, the authors have rewritten the sentence.
Yes, the failures were obtained in the heat affected zone.
17. Line 200-201 is repetition of line 197-199
Author’s Response: Thank you for this comment, the authors corrected the mistake.
18. Line 254: how the crack initiation looks like?
Author’s Response: The crack initiation sites are flat, without any noticeable defects for both the base
material and the FSW joints. These flat areas at the crack initiation sites suggest that fatigue cracking
was initiated by cyclic slip deformation due to local stress concentration.
19. What is difference between crack propagation and fast fracture region? Crack initiation is more influential? Why it was not discussed in the paper?
Author’s Response: The main aim of the article was not in to depth analyse the fracture mechanisms
but to compare the behaviour of FSW joints in terms of tensile and low cycle loading. In fatigue testing
of notched components, the location of crack initiation is crucial because the nucleation of the crack
strongly affects fatigue life. In our study it turned out that despite the presence of notches as remnants
of the FSW bonding, the cracks started from a smooth surface in the HAZ. Crack propagation region
includes stable crack growth, typical fatigue striations were observed in propagation region , as
shown in Fig. 12 or Fig. 13 in which secondary cracks along the fatigue striation can also be observed. Fast fracture region concerns a rapid crack growth in the final stage of the fatigue test.
20. Line 292: Indicate secondary crack by arrow
Author’s Response: Thank you for this comment, the authors have introduced a designation.
21. Conclusions need to be shorten and include only important obseration
Author’s Response: In the authors' opinion, the paragraph "Conclusions" is of an appropriate length
and contains observations based on the performed research.
22. Conclusions 5 is not clear (line 313-315)
Author’s Response: The authors have rewritten the sentence.
Best regards
Janusz Torzewski
Reviewer 2 Report
There is a lot of work behind the paper. The authors have done a good job on investigating the effect of friction stir welding on low cycle fatigue of AA5083 Al alloy. The fractography and presentation of the striations is especially well done. The results are interesting and the paper would make a good addition to Materials.
However, the main issue is with the language style, it is bad and often difficult to read, I recommend the authors deal with the issue.
There are also a few minor issues:
Please describe the H111 and H11 state in a bit more detail with a sentence or two.
Please do not use your own abbreviations for yield strength and ultimate tensile strength, Y and UT should be replaced with Rp02 and Rm, they are internationally used.
Also, when referring to Vickers measurements just use HV0.1, there is no need for the 0.98N load explanation.
Figure 2 is very informative, however if you would add "tensile sample" and "LCF sample" to the samples in the figure it would be even better.
Figure 7, the "material standards" are higher than described as the minimum in table 2, I suggest you only use the minimum values and present them as lines that go across all the columns, also use Rm, Rp02 and A.
Figure 9, make it more presentable, the values are all between 200 and 300, I don't see the need for a logarithmic scale on the amplitude
Author Response
Dear Sir or Madam
I am sending you the revised version of the paper that we submitted to your Journal “Materials”.
The authors acknowledge both the editor and the reviewer for carefully reading the paper. The comments are very interesting and useful and give a noticeable contribution to improve the paper.
We tried to satisfy all the suggestions of the reviewers.
In the manuscript, the main changes are highlighted in red. Please, note that, for the sake of clarity, the deleted text is not reported.
I am sending you the responses to reviewers
Comments and Suggestions for Authors
1. Please describe the H111 and H11 state in a bit more detail with a sentence or two.
Author’s Response: Thank you, the authors have introduced a short description of plastic working.
2. Please do not use your own abbreviations for yield strength and ultimate tensile strength, Y and
UT should be replaced with Rp02 and Rm, they are internationally used.
Author’s Response: Thank you again, the authors have introduced the proposed changes.
3. Also, when referring to Vickers measurements just use HV0.1, there is no need for the 0.98N
load explanation.
Author’s Response: The authors claim that such a description may appear in the section on Materials
and Methods.
4. Figure 2 is very informative, however if you would add "tensile sample" and "LCF sample" to the
samples in the figure it would be even better.
Author’s Response: Thank you, the authors have introduced the proposed changes
5. Figure 7, the "material standards" are higher than described as the minimum in table 2, I suggest
you only use the minimum values and present them as lines that go across all the columns, also
use Rm, Rp02 and A.
Author’s Response: Thank you, the authors took advantage of the reviewer's suggestion and introduced changes in Fig. 7.
6. Figure 9, make it more presentable, the values are all between 200 and 300, I don't see the need
for a logarithmic scale on the amplitude
Author’s Response: The authors fully agree with Reviewer, but the results in Fig. 9. were presented in
accordance with the rules for the presentation of low-cycle study results. In this case, the bilogarithmic scale is mandatory.
Best regards
Janusz Torzewski
Reviewer 3 Report
Please look at the comments within the enclosed paper;
Comments:
- space character
- It is not so easy to find suitable applications for LCF. Do you have another example in the shipbuilding industry, except slamming?
- 3. For FSW the material and geometry of the tool is also very important. Can you please insert the
basic statement to the tool. - This is a nice illustration. I am only surprised of the depcited positions of the specimen. Did you
really take the LCF specimen directly at the beginning of the weld? Regularly ther should be a little
offset from the plunge position due to the run in effect. - Did you grind the tensile test and fatigue specimen before testing or use them as welded?
- It would be better if you would mark of these pictures the area in figure 3.
- Also here please insert the lines in the same colors for the microhardness measurements in the cross
section image. - I do not udnerstand this well. In the diagramm I don´t see any decrease in elongation for the FSW
welds. They achieve higher values then the base material. - The effect of cyclic hardening appears a bit less prononunced for the loop 1 and loop 5 for the FSW
compared to the base material. Why? - Which phase?
- Please give an explanation( line 243), why
- Same fracture area as for the tensile testing?

Author Response
Dear Sir or Madam
I am sending you the revised version of the paper that we submitted to your Journal “Materials”.
The authors acknowledge both the editor and the reviewer for carefully reading the paper. The comments are very interesting and useful and give a noticeable contribution to improve the paper.
We tried to satisfy all the suggestions of the reviewers.
Comments and Suggestions for Authors
1. space character
Author’s Response: The authors checked the spacing throughout the article as suggested.
2. It is not so easy to find suitable applications for LCF. Do you have another example in the shipbuilding industry, except slamming?
Author’s Response: LCF simulates situations in which local high stresses generate permanent deformations particularly dangerous in the vicinity of joints, e.g. mooring, collision with an iceberg.
3. For FSW the material and geometry of the tool is also very important. Can you please insert the
basic statement to the tool.
Author’s Response: The authors introduced the sentence regarding the geometry of the tool used.
4. This is a nice illustration. I am only surprised of the depcited positions of the specimen. Did you
really take the LCF specimen directly at the beginning of the weld? Regularly ther should be a little
offset from the plunge position due to the run in effect.
Author’s Response: The authors thank you for your kind comment. The illustration shows schematically the position of the test specimens. The LCF specimens were cut out taking into account both the
beginning and the end of the weld.
5. Did you grind the tensile test and fatigue specimen before testing or use them as welded?
Author’s Response: Both types of samples (tensile and fatigue) were not ground before the test.
6. It would be better if you would mark of these pictures the area in figure 3.
Author’s Response: The authors fully agree with Reviewer, but in this case the pictures are on
different pages and it seems to us that the markings and description of the pictures are easy to identify.
7. Also here please insert the lines in the same colors for the microhardness measurements in the cross
section image.
Author’s Response: Thank you for this comment, the authors added the microhardness line
designation in photo 6 and an additional description in the article.
8. I do not udnerstand this well. In the diagramm I don´t see any decrease in elongation for the FSW
welds. They achieve higher values then the base material.
Author’s Response: The authors agree that this description may have been questionable. The data
concerned changes in strength parameters in relation to the base material of the tested batch. As
suggested by the second reviewer, the graph was corrected. On Fig. 7 dashed lines are used to mark
the minimum parameters resulting from the material standards contained in Table 1.2.
9. The effect of cyclic hardening appears a bit less prononunced for the loop 1 and loop 5 for the FSW
compared to the base material. Why?
Author’s Response: The base material is homogeneous, while in the FSW joint there are areas that
have been strongly deformed and subjected to additional thermal influence during the welding
process.
10. Which phase?
Author’s Response: The authors referred to the literature [26], where a detailed description can be
found. "This serrated flow has occurred due to the dynamic strain aging or the dynamic strain induced
evolution of the second phase (Al6Mn). The dislocation pinning solute drag mechanism, diffusion and
clustering of Mg were reported as the reason for the dynamic strain ageing."
11. Please give an explanation( line 243), why.
Author’s Response: The authors presented the justification of different durability of FSW samples
when describing fracture surfaces (paragraph 3.5).
12. Same fracture area as for the tensile testing?
Author’s Response: Yes, this was included in the conclusions as point 5.
Best regards
Janusz Torzewski
Round 2
Reviewer 1 Report
- What is “specimen surface” in abstract? Please specify (line 21)
- Some sentence still required reconstruction. For example
“The fatigue failure of FSW joints was observed to occur in the heat-affected zone. Fatigue crack propagation region was basically characterized by fatigue striations, together with some secondary cracks for all strain amplitudes and for both types of samples.”
“The boundary between TMAZ and SZ can be observed in Fig. 5a. The picture shows deformed, elongated grains present in TMAZ, which reflect the direction of material flow around the tool during the welding process. Affecting of the tool on the work piece in SZ results in the formation of ultrafine grain microstructure in this area due to the dynamic recrystallization phenomenon (Fig. 5b)”
“The wider the striation space is, the faster crack propagates described by Alatorre et al. [29].”
- Does the mountain map software give grain size automatically? If so what standard it uses for grain size measurement?
Author Response
Dear Reviewer,
The authors thank you for accurate comments and helpful suggestions.
In the manuscript, the main changes are highlighted in red. Please, note that, for the sake of clarity, the deleted text is not reported.
I am sending you the responses to your comments
1. What is “specimen surface” in abstract? Please specify (line 21)
Author’s Response: The sentence has been rebuilt and supplementary information has been introduced.
2. Some sentence still required reconstruction. For example
Author’s Response: Thank you for a comprehensive review. The authors re-checked the text as recommended and made changes to the specified sentences.
“The fatigue failure of FSW joints was observed to occur in the heat-affected zone. Fatigue
crack propagation region was basically characterized by fatigue striations, together with some
secondary cracks for all strain amplitudes and for both types of samples.”
For all tested strain amplitudes, the fatigue crack propagation region is characterized by the presence
of fatigue striation with secondary cracks.
“The boundary between TMAZ and SZ can be observed in Fig. 5a. The picture shows deformed, elongated grains present in TMAZ, which reflect the direction of material flow
around the tool during the welding process. Affecting of the tool on the work piece in SZ results in the formation of ultrafine grain microstructure in this area due to the dynamic recrystallization phenomenon (Fig. 5b)”
The boundary between TMAZ and SZ can be observed in Fig. 5a. The TMAZ is characterized by deformed, elongated grains reflecting the direction of material flow in the welding process. Affecting of
the tool on the workpiece results in the formation of ultrafine grain microstructure in the SZ due to the
dynamic recrystallization phenomenon (Fig. 5b).
“The wider the striation space is, the faster crack propagates described by Alatorre et al.
[29].”
Alatorre et al. reported that the wider striation is related to faster crack propagation [29].
3. Does the mountain map software give grain size automatically? If so what standard it
uses for grain size measurement?
Author’s Response: Mountains map 7 software is equipped with Grains & Particles Module. The
grains are detected by the binary segmentation algorithm and then subjected to the morphological
correction operator. Finally, the statistics for all measured grains together with histogram are obtained. Unfortunately, there is no information about the specific standard used by this software in
terms of grains measurements only that ISO 25178 is used for determining surface texture parameters.
For this reason, we did not give a specific value of grain size, only the range containing the most common, determined values.
Yours sincerely,
Janusz Torzewski

Reviewer 2 Report
I feel that the authors have followed the review and addressed all the issues. I think the manuscript is suitable for publication.
Author Response
Dear Reviewer
The authors thank you for accurate comments and helpful suggestions.
Yours sincerely
Janusz Torzewski
Reviewer 3 Report
Please see attached file for 2 new comments;
Overall, I think is is a really interesting work
Line 189: These results confirm the positive influence of grain refinement in
the SZ area observed in Fig. 4 and Fig. 5. on the strength
properties of aluminum alloy 5083 H111.
Line 242: The
observed effect of the increase in stress range under the
influence of strain for the AA5083 alloy can be
explained by the increase in dislocation density and the
mechanism of solid solution operation and the presence
of a fine second phase (Al6Mn) recognized in the
literature [26].
Author Response

(The authors gave the same response as above.)
